# Direct prediction of gas adsorption via spatial atom interaction learning

Jiyu Cui [1,6], Fang Wu [2,3,4,6], Wen Zhang [2,6], Lifeng Yang [1,3,6], Jianbo Hu [1,3], Yin Fang [2,3,5], Peng Ye [2,3,5], Qiang Zhang [2,3,5], Xian Suo [1,3], Yiming Mo [1,3], Xili Cui [1,3], Huajun Chen [2,3,5] ✉ & Huabin Xing [1,3] ✉

Physisorption relying on crystalline porous materials offers prospective avenues for sustainable separation processes, greenhouse gas capture, and energy storage. However, the lack of end-to-end deep learning model for adsorption prediction confines the rapid and precise screen of crystalline porous materials. Here, we present DeepSorption, a spatial atom interaction learning network that realizes accurate, fast, and direct structure-adsorption prediction with only information of atomic coordinate and chemical element types. The breakthrough in prediction is attributed to the awareness of global structure and local spatial atom interactions endowed by the developed Matformer, which provides the intuitive visualization of atomic-level thinking and executing trajectory in crystalline porous materials prediction. Complete adsorption curves prediction could be performed using DeepSorption with a higher accuracy than Grand canonical Monte Carlo simulation and other machine learning models, a 20-35% decline in the mean absolute error compared to graph neural network CGCNN and machine learning models based on descriptors. Since the established direct associations between raw structure and target functions are based on the understanding of the fundamental chemistry of interatomic interactions, the deep learning network is rationally universal in predicting the different physicochemical properties of various crystalline materials.

Physisorption based on porous materials offers cost- and energy-efficient alternatives toward promising solutions to global challenges in carbon dioxide ($CO_2$) capture[1,2], energy gas storage[3], separation[4–6] and etc, which consumes 10-15% of global energy consumption[7]. The breakthrough of related technologies lies in the design and screening of porous materials with specific adsorption properties, a critical characteristic that determines the functions of porous materials[8,9]. Crystalline porous materials, including metal-organic frameworks (MOFs)[10] or porous coordination polymers (PCPs)[11,12], covalent-organic frameworks (COFs)[13], and zeolites[14], can be rationally customized via the selective self-assembly of molecular building blocks[15], enabling the possibility of a bottom-up design of porous materials with envisaged functions. These materials have shown attractive potential in diverse fields, such as adsorption[16], membrane separation[17], and catalysis[18]. However, the discovery of porous materials is greatly hindered by the problems of long experimental times, high costs of conventional trial-

[1]Key Laboratory of Biomass Chemical Engineering of Ministry of Education, College of Chemical and Biological Engineering, Zhejiang University, 310012 Hangzhou, China. [2]College of Computer Science and Technology, Zhejiang University, 310027 Hangzhou, China. [3]Engineering Research Center of Functional Materials Intelligent Manufacturing of Zhejiang Province, ZJU-Hangzhou Global Scientific and Technological Innovation Center, 311215 Hangzhou, China. [4]School of Professional Studies, Columbia University, New York, NY 10027, USA. [5]Alibaba-Zhejiang University Joint Research Institute of Frontier Technologies, 310027 Hangzhou, China. [6]These authors contributed equally: Jiyu Cui, Fang Wu, Wen Zhang, Lifeng Yang. ✉e-mail: huajunsir@zju.edu.cn; xinghb@zju.edu.cn

and-error paradigms, and the limited efficiency of high-throughput simulation studies.

Machine learning affords a powerful approach for the rapid discovery of materials with desired adsorption properties by learning the knowledge of porous materials and their physisorption behaviors[19–22]. However, the accurate prediction of adsorption performance still remains a challenge due to the complex associations between raw material structures and functional properties that require machine learning models to understand the correlations among global atoms, local atoms with different element definitions[23–25]. Researchers have attempted to develop expert-engineered porous material descriptors that can maximally cover the key structural information to improve prediction accuracy[25–28]. However, since every piece of subtle structural information is crucial to the correct expression of adsorption properties, the intrinsic drawback of raw structural information loss and high computational cost during the descriptors generation and processing inevitably cause error propagation[21,27,29]. Even for the most commonly acknowledged structural descriptors, for example, largest cavity diameter (LCD), the Pearson correlation coefficient between the LCD and $CO_2$ adsorption capacity is only −0.14, via the preliminary data analysis of the gas adsorption performance of porous materials (Fig. S1). End-to-end prediction is favorable for maintaining complete raw structural information, and it has great potential for accurate prediction[30,31]. However, three daunting challenges have yet to be

addressed to realize efficient direct structure-adsorption prediction: (i) advanced models are needed for translating and transferring complete raw structural information, including both chemical element knowledge and spatial atomic arrangement; (ii) atomic-level information needs to be exchanged for the accurate cognition of spatial atomic interactions and good interpretability of the model; (iii) the efficient utilization of knowledge from field experts is required to solve the data hungry problem in solely data-driven deep learning models.

Here DeepSorption, a data-driven network with a built-in expert knowledge co-learning (KCL) module, was designed for fast and end-to-end predictions directly from the coordinates and elements of atoms to the adsorption properties of porous materials (Fig. 1) and it achieved the best prediction results on multiple data sets. The distinctive architecture of the network lies in the developed Matformer model that serves as a high-fidelity interpretation of the overall structural information of porous materials, including the atomic spatial arrangement and chemical element information. Moreover, the Multi-scale Atom-attention (MSA) mechanism within the model realizes the accurate, efficient cognition of interactions between atoms at different scales, and enables the visualization of the potential atomic interactions hidden in the encoding layers. The KCL module reduces the reliance of the end-to-end network on massive training data, and is beneficial for improving the accuracy of adsorption property prediction. DeepSorption far outperforms other available networks in

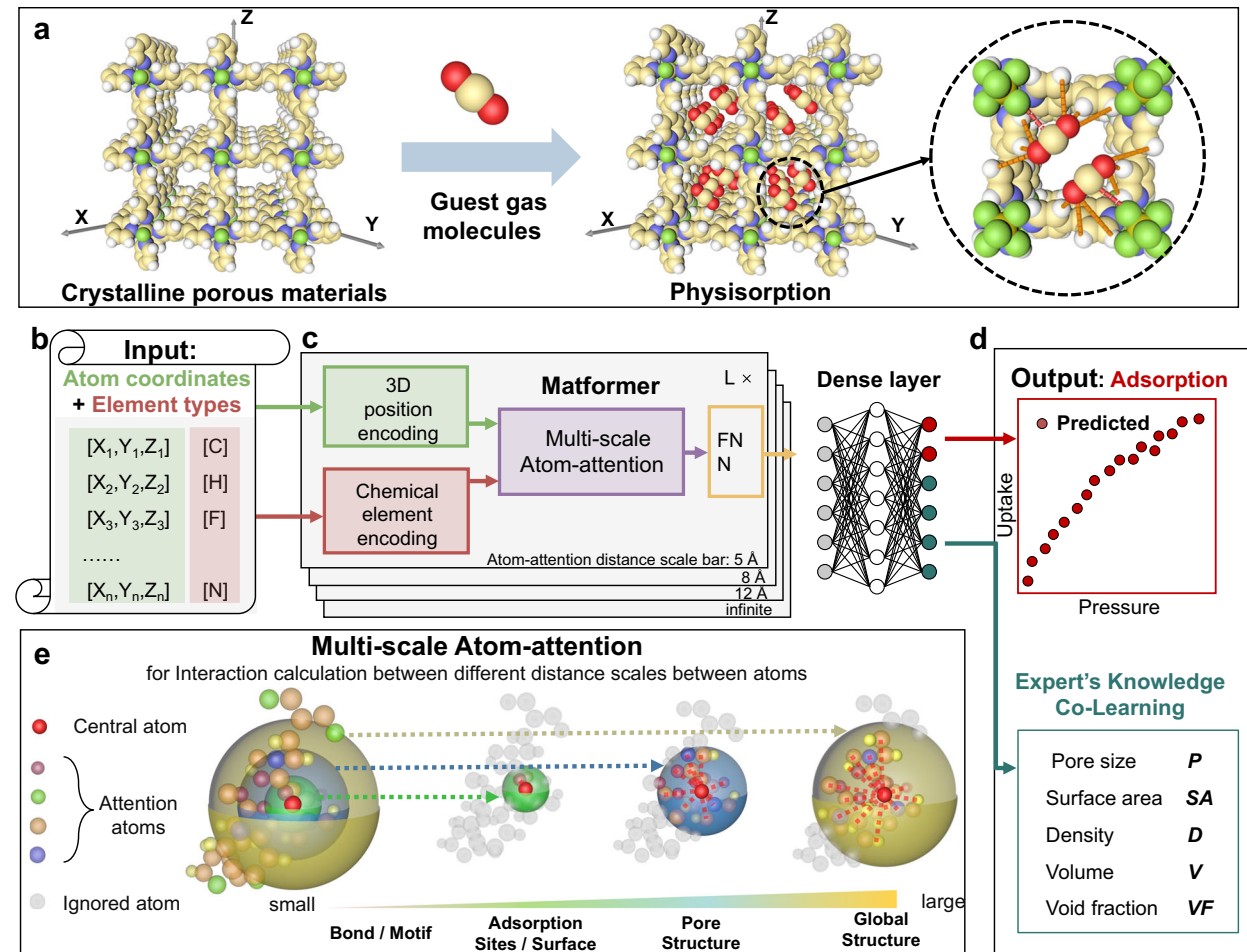

**Fig. 1 | Representation of DeepSorption network. a** The scheme of crystalline porous materials (SIFSIX-1-Cu), guest molecules (carbon dioxide) and physisorption process. **b** The inputs of DeepSorption, including atom coordinates and element types. **c** The architecture of Matformer, including 3D position encoding layer, Chemical element encoding layer, Multi-scale Atom-attention layer, and feed-forward neural network (FNN). **d** The outputs of DeepSorption, including gas adsorption isotherms and co-learning expert's knowledge. **e** The scheme of Multi-scale Atom-attention for calculating the interaction between atom pairs in different distance scales.

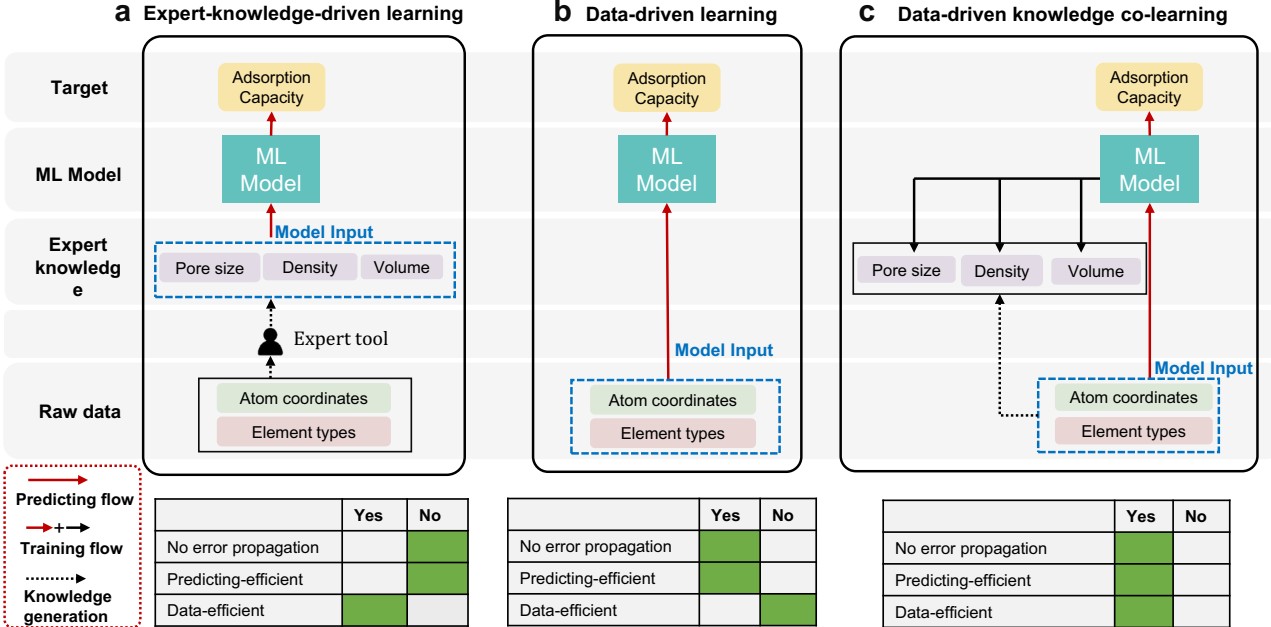

**Fig. 2 | Different machine learning (ML) models. a** The scheme of Expert-knowledge-driven learning model. The raw data is transformed into corresponding descriptors via expert knowledge, serving as the input of the subsequent machine learning model. **b** The scheme of Data-driven learning model. Raw data is directly as the input of the model to predict the adsorption capacity. **c** The scheme of Data-driven knowledge co-learning model. Raw data is directly as the input of the model to predict the adsorption capacity, and the expert knowledge is used as auxiliary prediction tasks.

adsorption uptake predictions with lowest root mean squared errors (RMSE) and highest correlation coefficients both in $CO_2$ and acetylene ($C_2H_2$) prediction, which is essential for identifying efficient adsorbents for $CO_2$ capture and $C_2H_2$ separation.

## Results

### Designing and constructing the deep learning network
Crystalline porous materials could exhibit predesigned skeletons and nanopores through the atomically precise integration of organic units, inorganic units or combinations. Physisorption reveals the different priority locations of guest molecules within nanopores that are mainly governed by interatomic interactions by exploring the potential adsorption sites and space (Fig. 1a). Here, the original data of crystalline materials, including atom coordinates and element types, are directly used as the input of DeepSorption (Fig. 1b), preventing information loss caused by using processed expert knowledge descriptors as input. Given that the characteristic of the process by which atoms form frames is inherently similar to that of natural language, different arranged atoms (words) construct the specifically defined framework (sentence), a home-made Matformer inspired by natural language processing[32] is employed to process crystalline material data. The encoding of 3D atomic coordinates and the corresponding element types utilizes 3D position encoding block and Chemical element encoding block (Fig.1c). 3D position encoding is an absolute position encoding approach that endows the model with both local and global structure-aware abilities. Chemical element encoding block is initialized via chemical element knowledge graph (Fig. S2), which is built from periodic table and summarizes the most fundamental chemical properties of elements and microscopic associations among elements.

The key innovation in Matformer lies in Multi-scale Atom-attention (MSA) for understanding the interactions between different defined atoms in a spatial arrangement (Fig. 1e). Through exchanging information between atom pairs in different distances, MSA facilitates Matformer with the intuition to judge the interatomic interaction at different scales. In detail, MSA computes the atomic distance based on the input atomic coordinates, and the contribution of the atomic

element type is simultaneously considered. The hyper-parameter attention distance bars are 5 Å, 8 Å, 12 Å and infinite, corresponding to bond/motif detection, adsorption site and surface decisions, pore structure detection, and global structure awareness, respectively (Fig. 1e). Interactions between atom pairs at different attention distance scales could be pointedly processed by the appointed units in Matformer, and the processed information, such as key atoms/space, would be integrated to reveal physisorption behavior of guest molecules. With regard to the data-hungry drawbacks occurred in direct data-driven learning, the strategy of knowledge co-learning (KCL) is employed (Fig. 2), and the descriptors of crystalline porous materials are set as auxiliary tasks (Fig. 1d). The results show that KCL could facilitates the convergence of the model in the structure-adsorption space establishment assisted by the expert knowledge derived from the auxiliary tasks, which is beneficial for improving the prediction accuracy of adsorption properties. It is noteworthy that the output of expert knowledge is only needed during model training, but leaving no interference with the prediction process, guaranteeing the rapid prediction speed. DeepSorption well inherits the advantages of time efficiency, decreased error propagation of end-to-end method, and the data efficiency of the expert-knowledge-driven learning method (Fig. 2).

### Model performance and validation
To train DeepSorption (Matformer+KCL) deep learning network, the data of crystalline materials collected in the CoREMOF database[33] (including over 11,000 MOFs and 77 kinds of elements) and hMOF database[34] (including over 300,000 MOFs and 16 kinds of elements) were used. $CO_2$ capture is crucial to alleviate global warming, facilitating carbon conversion and utilization, and developing efficient carbon capture technologies are subjected to the discovery of adsorbents with high $CO_2$ and low $N_2$ capacity[34,35].

Thus, $CO_2$ and $N_2$ are used as tested gases in this work, and the predicted gas uptake is consistent with the true values in all tasks, including CoREMOF-$CO_2$ (Fig. 3a), hMOF-$CO_2$ (Fig. 3b) and hMOF-$N_2$ (Fig. S3). DeepSorption shows much smaller and more distributed

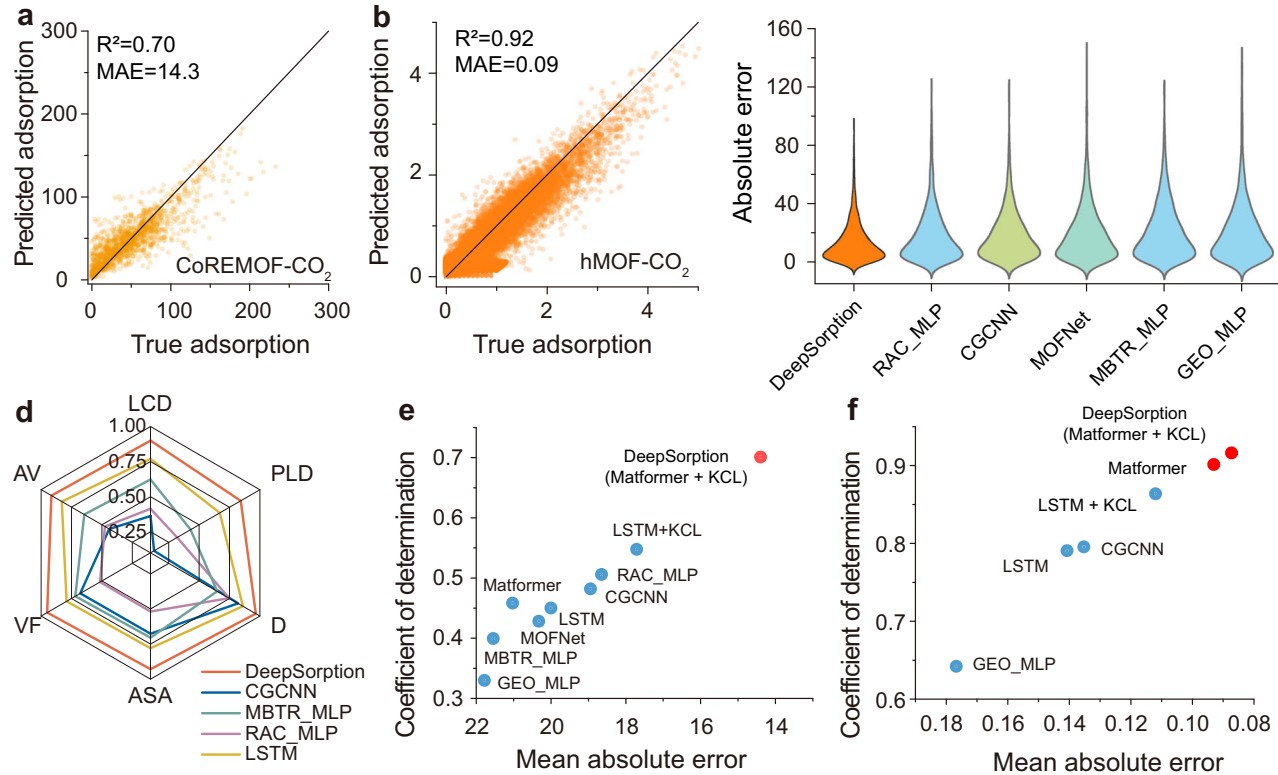

**Fig. 3 | Prediction performance of DeepSorption on CoREMOF and hMOF datasets. a, b** The correlations between true adsorption uptake and predicted adsorption uptake on test sets implemented by DeepSorption network on CoREMOF-$CO_2$ (**a**), hMOF-$CO_2$ (**b**) tasks. **c** The distribution of absolute errors on test sets between true adsorption uptake and predicted adsorption uptake on CoREMOF-$CO_2$ based on different models, including DeepSorption (Matformer +KCL), RAC_MLP (Multilayer perceptron based on RACs descriptors), CGCNN (Crystal Graph Convolutional Neural Network[47]), MOFNet[19], MBTR_MLP (Multilayer perceptron based on MBTR descriptors), GEO_MLP (Multilayer perceptron based on geometric structure descriptors). **d** The coefficient of determination $R^2$ on test sets of co-learning knowledge, including LCD (largest cavity diameter), PLD (pore limiting diameter), D (density), ASA (accessible surface area), VF (void fraction), AV (accessible volume) on CoREMOF-$CO_2$ using DeepSorption, CGCNN, MBTR_MLP, RAC_MLP and LSTM (Long Short-Term Memory[48]) models. **e, f** The coefficient of determination $R^2$ and mean absolute errors (MAE) of different models on CoREMOF-$CO_2$ (**e**) and hMOF-$CO_2$ (**f**) tasks on test sets.

centralized absolute errors on the CoREMOF-$CO_2$ task than the other models, and the mean absolute error (MAE) of DeepSorption (14.39 $cm^3$ $g^{-1}$) decreased by 23–52% compared with those of Long Short-Term Memory (LSTM: 20.00 $cm^3$ $g^{-1}$), Crystal Graph Convolutional Neural Network (CGCNN: 18.94 $cm^3$ $g^{-1}$), Expert-knowledge-driven learning models (GEO_MLP: 21.78 $cm^3$ $g^{-1}$, RAC_MLP: 18.64 $cm^3$ $g^{-1}$, MBTR_MLP: 21.54 $cm^3$ $g^{-1}$, SOAP_MLP: 30.05 $cm^3$ $g^{-1}$) (Fig. 3c). Moreover, higher coefficient of determination ($R^2$) values and lower RMSE of co-learning knowledge tasks of LCD, pore limiting diameter (PLD), density (D), accessible surface area (ASA), void fraction (VF), accessible volume (AV) are also realized (Fig. 3d and Fig. S3). The improved prediction accuracy of physisorption associated parameters is attributed to the global structure awareness ability of Matformer. On average, for the CoREMOF-$CO_2$, hMOF-$CO_2$ and hMOF-$N_2$ tasks, KCL contributes to a 13% decrease in RMSE and an 18% increase in $R^2$. These results not only indicate that DeepSorption, like human scientists, could well learn and utilize expert knowledge, but also explain why data-driven knowledge co-leaning models can achieve a better learning effect compared to the solely data-driven learning methods. In contrast to the classic EKDL model and graph neural networks, DeepSorption always exhibits the highest $R^2$ value and the lowest MAE value (Fig. 3e, f). The $R^2$ value of the predicted $CO_2$ adsorption uptake in the CoREMOF dataset is over 0.70, which is increased by 38–113% than those of EKDL models (Fig. 3e and Table S1). Compared with the graph neural network CGCNN, the performance is also significantly improved, from 0.48 to 0.70 of $R^2$ (Fig. 3e). The great advancement in prediction accuracy of DeepSorption is attributed to the unique

advantages of the designed network that realizes the complete interpretation of original structural information and the comprehensive understanding of spatial atom interactions. For CGCNN, this insufficiency was attributed to the poor global structure awareness characteristic of graph neural network itself. As shown in Fig. S15, the strategy of graph neural network is to strengthen the information of atomic elements by calculating the neighbor atoms via using the spatial coordinate information. However, the information of atomic spatial coordinate would be lost in the subsequent information interaction process, which is not conducive to predict adsorption properties that are sensitive to the global spatial structure information.

For further clarifications on the KCL (knowledge co-learning) procedure, we tested using only a subset of descriptors as KCL and analyzed the effect enhancement of different subsets (Table S10). According to the correlation coefficient between the structure descriptors and $CO_2$ adsorption, we selected the following sets of tasks ([AD], [AD, LCD], [AD, LCD, PLD], [AD, LCD, PLD, AV], [AD, LCD, PLD, AV, ASA], [AD, LCD, PLD, AV, ASA, D], [AD, LCD, PLD, AV, ASA, D, VF]). And we found that the model achieved the best prediction results when using four descriptors (LCD, PLD, AV and ASA), slightly higher than the prediction results when using all six descriptors ($R^2$: 0.708 vs 0.701, MSE: 419 vs 429). We also found that the difference between results of using no auxiliary tasks and incorporating any auxiliary tasks was huge. Through the following comparative experiments, it is speculated that the difference is caused by the fact that the physical structure descriptors as auxiliary tasks can activate the position-encoding module which takes absolute coordinates as the input.

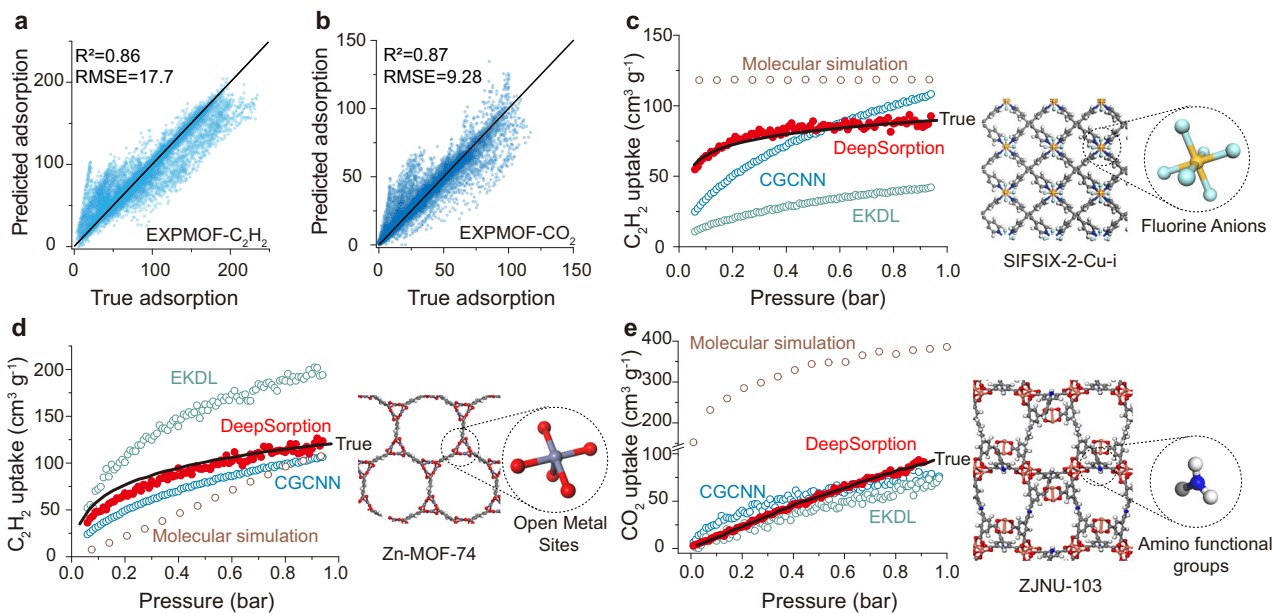

**Fig. 4 | Prediction performance of DeepSorption on experimental dataset (EXP-MOF). a, b** The correlations between true adsorption uptake and predicted adsorption uptake from DeepSorption network on EXPMOF-$C_2H_2$ (**a**) and EXPMOF-$CO_2$ (**b**) tasks on test sets through leave-one-out validation. **c–e** The experimental and predicted adsorption isotherms via different machine learning algorithms, red for DeepSorption, green for EKDL (Expert-knowledge-driven learning model), blue for CGCNN (Crystal Graph Convolutional Neural Network) and yellow for Grand canonical Monte Carlo molecular simulation of SIFSIX-2-Cu-i with $SiF_6^{2-}$ anions (**c**), Zn-MOF-74 with open metal sites (**d**), and ZJNU−103 (**e**) with amino functional groups.

To examine the sensitivity of DeepSorption toward the training-validation-test set random split of the prediction results, we performed 10 different random divisions of the $CO_2$ adsorption data in CoREMOF dataset (Table S12) and found that the results are less affected by the division of the dataset, with $R^2$ between 0.672– 0.712 and MAE between 13.598–15.150 $cm^3\,g^{-1}$, proving the robustness of the model.

The ability to predict out-of-distribution data is challenging and is an important indicator of the model's generalizability. Elements that occur less than one in a thousand times are defined as rare elements (As, Rh, Sb, Te, Ir, Pb, Np and Pu). As shown in Fig. S29 we have compared the predicted and true values of the $CO_2$ adsorption capacity of the MOF containing the rare elements in the test set. DeepSorption still showed the best prediction performance ($R^2$: 0.28), better than the other comparison models, CGCNN ($R^2$: -0.23) RAC_MLP ($R^2$: -0.15) and GEO_MLP ($R^2$: 0.09). We attribute this good out-of-distribution prediction to the use of the Chemical Element Knowledge Graph for element coding in MatFormer. The Chemical Element Knowledge Graph coding method gives each element a vector representation containing chemical element information by learning and correlating the inter-relationships between the properties of the elements. The vector representations are learnt based on Chemical Element Knowledge Graph and is independent to the training data of gas adsorption prediction, so that even if materials contain elements in the test data that do not appear in the training data or appear only a few times, the model can still give relatively accurate adsorption predictions since the out-of-distribution elements also have information-rich vector representations.

We further examined the performance of DeepSorption model at a wide range of conditions, including carbon dioxide (2.5 bar and 298 K), methane (35 bar and 298 K) and hydrogen (100 bar and 77 K) adsorption capacity prediction tasks. As presented in Fig. S30 and Table S13, DeepSorption model still showed better prediction performance compared with other models, and we also found that $R^2$ of adsorption prediction tasks of high pressure was generally higher than those of low pressure. The $R^2$ of the three tasks of $CO_2$ (2.5 bar and

298 K), $CH_4$ (35 bar and 298 K) and $H_2$ (100 bar and 77 K) adsorption reached 0.96, 0.98 and 0.99 respectively via DeepSorption models, which may attribute to the fact that the adsorption capacity is mainly determined by the surface area and pore volume of the material under high pressure. This phenomenon can also be drawn from the better prediction effect of GEO_MLP on $H_2$ (100 bar and 77 K) task ($R^2$: 0.994), than CGCNN ($R^2$: 0.872), since the latter is not good at capturing the overall spatial structure of materials.

The superiority of DeepSorption is further validated on the collected experimental $C_2H_2$ (EXPMOF-$C_2H_2$) as well as $CO_2$ (EXPMOF-$CO_2$) adsorption isotherms. $C_2H_2$ adsorption is essential to its safe storage, as well as the key technology for the production of polymer-grade ethylene[36]. A low RMSE is obtained using leave-one-out validation on the experimental data, where $R^2$ reaches 0.86 for EXPMOF-$C_2H_2$ and 0.87 for EXPMOF-$CO_2$ (Fig. 4a and b). Given the easy occurrence of severe deviations in the adsorption property prediction of porous materials with strong interaction sites, three tasks that involves typical strong polar sites, SIFSIX-2-Cu-i[37] with anion sites, Zn-MOF-74[38] with open metal sites, and ZJNU−103 with amino functional groups (Fig. 4c–e) are performed. It is noteworthy that DeepSorption still shows highly consistent values with the experimental ones, and moreover the prediction of $C_2H_2$ and $CO_2$ adsorption isotherms from 0 to 1 bar could be completed within seconds using only one 3090 RTX GPU. Despite a longer computing time (tens of hours), the adsorption prediction performance of molecular simulation is still unsatisfactory in the low pressure adsorption zone (Fig. 4c–e). DeepSorption also well outperforms other exiting learning approaches for adsorption properties prediction, and its improvement is fully demonstrated in the case of known SIFSIX-2-Cu-i featured with steep $C_2H_2$ adsorption curve (Fig. 4c). Since the adsorption isotherms of $C_2H_2$ and other strong polar gases under low pressure are incline to be governed by the host-guest interactions, such as hydrogen bonding interactions between $SiF_6^{2-}$ and $C_2H_2$ in SIFSIX-2-Cu-i, instead of the pore volume and surface area, its accurate predictions require the model to gain insight into spatial interaction learning. By contrast, CGCNN and EKDL that are

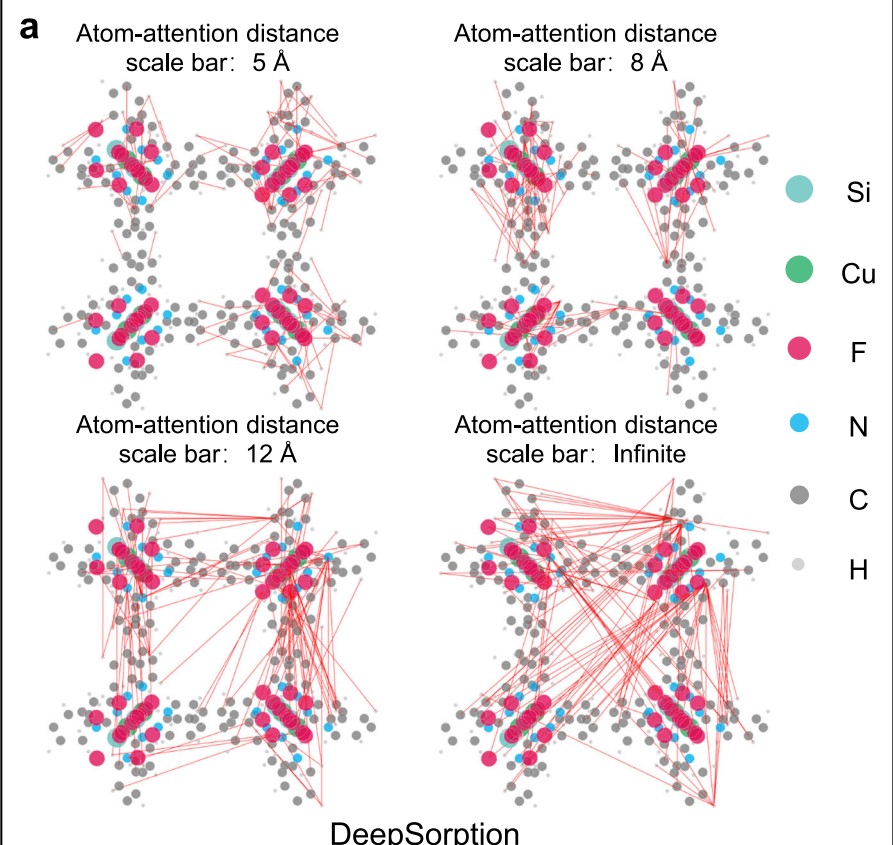

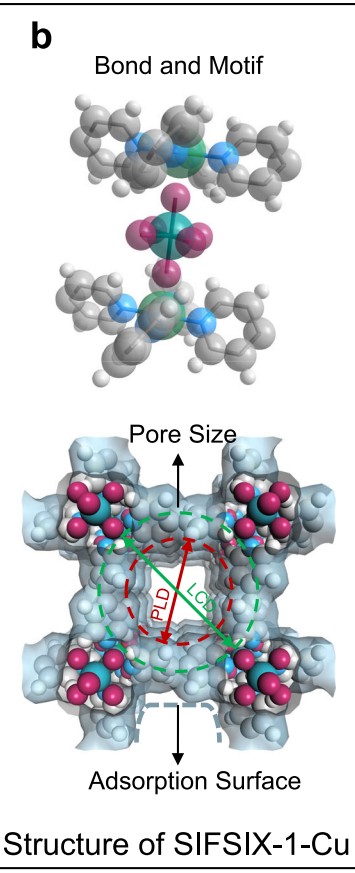

**Fig. 5 | 3D attention visualization. a** The 3D attention maps of SIFSIX−1-Cu at different attention distance scale bars, including 5 Å, 8 Å, 12 Å, and infinite. **b** The structure of SIFSIX−1-Cu, including bond, motif, pore size and adsorption surface (LCD: largest cavity diameter, PLD: pore limiting diameter). (Color code: C, Gray-50%; Si, Cyan; H, Gray-25%; N, Blue; Cu, Green; F, Pink; The attention between atom pairs, Red).

insensitive to either spatial information or elemental chemical information, are incapable to well evaluate interatomic interactions and fail to accurately predict steep adsorption curves. The accurate and highly-efficient prediction based on DeepSorption is believed to largely accelerate the discovery of crystalline porous materials with specific adsorption properties.

## Chemical insights at the atomic level

The interpretability of models has been a long-term concern in the field of 'AI for science'. MSA mechanism endows DeepSorption with the logical thinking of atomic-level interactions, enabling the intuitive visualization of the execution trajectory, which is essential to deepen the understand about the learning process of the network. SIFSIX−1-Cu, one of the benchmark $C_2H_2$ adsorbents[36], serves as an example (Fig. 5a). The relationship between priority attention atoms is presented in Fig. 5, and the high priority atom pairs in different attention distance scale are highlighted with lines. At the 5 Å scale bar, Deep-Sorption mainly focuses on the interactions between neighboring atoms, such as Cu⋯N, N⋯C and F⋯H, to search for the important motif and bond (Fig. 5a). At 8 Å scale bar, the local structural characteristics of the material are focused on by evaluating the interactions between F, Si atoms on the SIFSIX$^{2-}$ anions and H, C atoms on the organic ligands, to explore the potential adsorption sites and pore surface for gas accommodation. The deep insight into MOFs with different structures also indicates that the adjacent pairs of atoms with large electronegativity differences as the potential polar adsorption sites are preferentially concerned within small scale bars, such as Cu⋯O in HKUST-1[39] and Zn⋯O in UTSA-74[40] (Fig. S31 and S32). At 12 Å scale bar, the interactions between atoms on the surface of the pore channel

begin to be considered, which implies that the model tries to learn and calculate expert knowledge information, such as pore diameter, pore volume and surface area (Fig. 5b). As revealed in SIFSIX-1-Cu, Deep-Sorption not only focuses on the C⋯C and C⋯H atom pairs on the sides of square channels for PLD measurement, but also the interaction of F⋯C and F⋯H atom pairs on the diagonal of square channels to measure LCD. At the infinite scale bar, interactions between heavy atoms Si, Cu and distant atoms are highlighted, which are beneficial for model to understand and extract the overall topology and global structure information of the crystalline materials. In addition, in MOFs with open metal sites, such as MFM-188[41], MOF-505[42], HKUST-1[39] and UTSA-74[40], (Figs. S33 and S34), most of the attentioned atom pairs are related to unsaturated metal sites Cu or Zn at all atom-attention distance scale bars, which verified that DeepSorption has possessed the ability to judge the critical important binding sites to give the accurate gas adsorption isotherms.

## Discussion

DeepSorption presents the spatial atom interaction learning network that realizes the accurate and fast prediction of complex adsorption properties of crystalline porous materials with benchmark prediction accuracy. Benefiting from the Multi-scale Atom-attention mechanism, DeepSorption is able to perform an accurate evaluation of interactions between atoms to achieve physisorption behavior prediction and offer an intuitive visualization of the thinking and executing trajectory which has not been realized in the existing networks for adsorption prediction. The remarkable advancement in the prediction of complex physicochemical properties highlights the importance of the global structure awareness, the coupling transfer and interaction of the

atomic-level spatial structure information and the chemical element information. The spatial atom interaction learning network reveals the intrinsic chemistry that determines the expressed function of crystalline porous materials, and is a promising powerful tool for the prediction of various physicochemical and surface properties of other crystalline materials with accessible atomic coordinates, such as perovskite and crystalline catalysts.

## Methods

### DeepSorption network

DeepSorption network is mainly composed of Matformer and KCL modules. During training of DeepSorption network, the cartesian three-dimensional coordinates and corresponding element types of atoms in crystalline porous materials are input. In order to assign a representation to each atom in crystalline porous materials, three-dimensional coordinate information and element type information of atoms are first encoded through 3D position encoding layer and Chemical element encoding layer respectively, and the spatial and element information are then added up to get the atomic representation. The obtained atomic representation in the crystalline porous materials are transferred and interacted among atoms through the Multi-scale Atom-attention layer (MSA) in Matformer model. After the computation of Matformer module with six layers, DeepSorption outputs the predicted value of adsorption capacity as the target task and expert knowledge (largest cavity diameter, pore limiting diameter, density, accessible surface area, void fraction, accessible volume) as the auxiliary tasks simultaneously based on the KCL module. More details of DeepSorption network are available in supporting information.

### MSA

Multi-scale Atom-attention (MSA), a scale-aware multi-head attention mechanism (Fig. S4), is designed to recognize the interactions between atoms at different scales. Its input is a sequence of atom vectors, and the output is a sequence of updated atom vectors in the same order as the input. For each attention head, we assign a visible distance $\tau_s$ to make atoms within $\tau_s$ visible to each other during attention calculation in this head. With the input atom representation sequence $(x_1, x_2, x_3, \ldots, x_i, \ldots, x_N)$, each head of MSA first generates a key, value and query based on each atom vector $x_i$:

$$q_i = f_Q(x_i) \tag{1}$$

$$k_i = f_K(x_i) \tag{2}$$

$$v_i = f_V(v_i) \tag{3}$$

where $q_i, k_i, v_i$ are query, key and value vector with dimension $d_k$. Then an attention score is calculated based on the similarity between query $q_i$ to key of atoms whose distance to atom $i$ is within $\tau_s$. Specifically, the attention that the atom $i$ pays to $j$ can be formulated as:

$$a_{ij}^{\tau_s} = \frac{q_i k_j^T \cdot 1_{\{d_{ij} < \tau_s\}}}{\sqrt{d_k}} \tag{4}$$

where $d_{ij} = ||p_i - p_j||_2$ is the Euclidean distance between atom $i$ and $j$, $1_{\{d_{ij} < \tau_s\}}$ is the indicator function that makes the score between two atoms beyond distance $\tau_s$ to 0, and $\frac{1}{\sqrt{d_k}}$ is a scaling factor. The output vector of atom $i$ at this attention head is:

$$z_i^{\tau_s} = \sum_{j=1}^{N} \sigma(a_{ij}^{\tau_s}) v_j \tag{5}$$

here $\sigma$ denotes the softmax function. For each attention head, we specify distinct distances to enable the model to capture knowledge at different scales. Then vectors of atom $i$ from different heads are concatenated resulting a multi-scale vector $z_i$, followed by a feed-forward network to map it to dimension $d_{model}$.

### KCL

Knowledge co-learning (KCL), a module of DeepSorption network, is utilized to guide the model to better and faster converge during training by learning to predict target tasks and auxiliary tasks closely related to target tasks synchronously. The selection of auxiliary tasks commonly depends on the expert knowledge in the field. It is worth mentioning that the expert knowledge only needs to be used as a data label during model training. The prediction of crystalline porous adsorbents does not require expert knowledge, only the coordinates and corresponding element types of atoms in crystalline porous adsorbents are required as input (Fig. 2c). In this way, the simplicity and speed of the prediction process will not be damaged. Expert knowledge such as pore size and pore volume used in this study can be obtained by high-throughput calculation using automated high-throughput analysis software Zeo++[43]. Zeo++ uses crystal structure information files of crystalline porous materials as input, which can calculate the expert knowledge of materials at high throughput without manual annotation by chemical experts. Zeo++ are based on the Voronoi decomposition, which for a given arrangement of atoms in a periodic domain provides a graph representation of the void space[44]. After having the representations of crystalline porous materials $\{z_i\}_{i=1,\ldots,N}$, we feed them into a fully connected layer to conduct prediction.

### Training details

To train DeepSorption, we use the crystalline materials collected in the CoREMOF, hMOF, EXPMOF-CO$_2$ and EXPMOF-C$_2$H$_2$ datasets. We split the CoREMOF and hMOF datasets with a ratio for train/validation/test as 0.7:0.15:0.15. For CoREMOF, we simultaneously predict the following 7 targets: LCD, PLD, D, ASA, VF, AV and AD$_{CO2}$ (adsorption uptake of CO$_2$). For hMOF, the following 7 targets are predicted at the same time: AD$_{CO2}$, AD$_{N2}$ (adsorption uptake of N$_2$), LCD, PLD, D, VF and ASA. We use leave-one-out method to evaluate the performance of our model in EXPMOF-CO$_2$ and EXPMOF-C$_2$H$_2$ datasets. In details, when predicting the adsorption performance of material X (X refers to any material in the EXPMOF database) in the test set, any adsorption data of material X will not appear in the training set for model training. For EXPMOF-CO$_2$, we predict the following targets at the same time: adsorption uptakes of CO$_2$ at different pressures (0.01–0.92 bar, spaced at 0.1 bar), LCD, PLD, D, ASA and AV. For EXPMOF-C$_2$H$_2$, we predict the following targets at the same time: adsorption uptakes of C$_2$H$_2$ at different pressures (0.06–0.95 bar, spaced at 0.1 bar), LCD, PLD, D, ASA and AV. In details DeepSorption is trained to minimize the MSE loss, which is the mean of the squared errors between true and predicted values on training data. We evaluate the performance of our model on test dataset.

### 3D attention visualization of DeepSorption

With the help of the Multi-scale Atom-attention mechanism, the interpretability at multiple scales is clearly presented in this study, weights are extracted from Multi-scale Atom-attention layers. Atom pair interactions with higher weight ranking are displayed by connecting lines to show the atomic interactions that the model pays more attention to in the predicting process. Taking 5 Å distance scale bar as an example, we first draw the structure of crystalline porous material by using atomic coordinate information and chemical element type information. Then, the Atom-attention weight parameters for each atom pair of Matformer with distance bar of 5 Å are extracted and sorted from largest to smallest. The top 20 atomic pairs in weight

order of each layer of Matformer in the distance bar of 5 Å are highlighted with red lines.

## Dataset

CoREMOF, hMOF and EXPMOF datasets were used in this study. hMOF dataset includes over 300,000 hypothetical MOFs containing 16 elements, which are built with rigid organic and inorganic struts called secondary building units (SBU)s. hMOF dataset includes 3D Cartesian coordinates and the corresponding element types of MOFs, and corresponding adsorption performances of carbon dioxide ($CO_2$), nitrogen ($N_2$) determined by Grand canonical Monte Carlo (GCMC) simulations, and their corresponding expert knowledge, including LCD, PLD, D, ASA, and VF. CoREMOF (Computation-Ready, Experimental MOF[33]) dataset includes over 11,000 computation-ready, experimental three-dimensional metal-organic frameworks (MOFs) that contains more than 77 elements. The dataset contains 3D Cartesian coordinates and the corresponding element types of MOFs, and the corresponding adsorption performance of carbon dioxide ($CO_2$) determined by Grand canonical Monte Carlo (GCMC) simulations[45], and their corresponding expert knowledge, including LCD, PLD, D, ASA, VF and AV (Fig. S3). It is worth mentioning that the expert knowledge is only needed during training process, but is not required in the prediction of crystalline porous adsorbents. The EXPMOF dataset constructed by ourselves is composed of EXPMOF-$CO_2$ and EXPMOF-$C_2H_2$. The adsorption data of EXPMOF dataset is from experiments. The 3D Cartesian coordinates and the corresponding element types of crystals are collected from reported literature or our lab. EXPMOF-$CO_2$ contains 112 data, and EXPMOF-$C_2H_2$ contains 140 data. In more details, adsorption isotherms are extracted from the figures of literature and then are interpolated for data alignment. The expert knowledge of MOFs, including LCD, PLD, D, ASA, and AV, are calculated using Zeo + + programs.

## Full algorithm details

Full explanations and details of deep learning model algorithm and hyper-parameter details (including Matformer and LSTM) are available in supporting information.

## Reporting summary

Further information on research design is available in the Nature Portfolio Reporting Summary linked to this article.

## Data availability

All data supporting the findings of this study are available within the main text and the Supplementary Information. The data that support the findings of this study are available in https://doi.org/10.5281/zenodo.7699719[46]. Source data are provided with this paper.

## Code availability

The code repository is stored at https://doi.org/10.5281/zenodo.7699719[46] and https://github.com/DeepSorption/DeepSorption1.0.

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

## Acknowledgements
This work was financially supported by the National Natural Science Foundation of China 22122811 (X.C.), 22227812 (H.X.), and 21938011 (H.X.).

## Author contributions
H.X., H.C., and J.C. conceived of the project. J.C., L.Y., J.H., X.S. collected and cleaned the dataset. F.W., J.C., P.Y. and W.Z. designed the models. F.W., P.Y., Y.F. and J.C. ran the experiments. Y.M., Q.Z., X.C., X.S. and L.Y. analysis the results and provided advice. H.X., H.C. and W.Z. directed and managed the project. J.C. and L.Y. wrote the paper with help and feedback from H.X., H.C., W.Z., Y.M., Q.Z., X.C., X.S.

## Competing interests
The authors declare no competing interests.
