## [Peer Review File · Nature Communications]

Direct prediction of gas adsorption via spatial atom interaction learningREVIEWER COMMENTS

Reviewer #1 (Remarks to the Author):

In the present article Cui et al., presented an end-to-end deep learning model for accurate predictions of gas adsorption in nanoporous materials. According to the proposed approach predictive models of high accuracy have been developed requiring as an input only the structure of the material, namely the atomic coordinates of the chemical elements. It is claimed that the model outperforms any other model.

It is sad to say that after reading a few times the present manuscript, I still have not a safe opinion for the value of the present work. In particular the way that the predictive models were developed and evaluated still remains unclear to me despite the fact that several information was provided by the authors. More importantly, I am not sure what is the range of the applicability of the models developed.

Below are some of my concerns:

*) Why CO₂ and C₂H₂ adsorption were studied only at such low pressures (up to 1 bar)? There are freely available several datasets with the materials CO₂ capacities computed at higher pressures. Also, there are several datasets for different gases and at different thermodynamic conditions which could be used by the authors for the evaluation of the approach. Although it is explained the practical value of predicting CO₂ and C₂H₂ adsorption at the specific conditions, for the purpose of assessing the performance of a model additional results should be presented.

It is known that from the physical point of view, there are different parameters that determine the adsorption of the gas at different thermodynamic conditions. For example, at low pressures the materials' composition is the most important factor, since the amount of the adsorbed gas is mainly determined by the strength of its interactions with the material. On the other hand, at high pressures the available volume is the most decisive factor. Examining the performance at a wide range of conditions is important for a thorough evaluation of the present model

*) Some chemical elements (metals) out of the 77 in the Core MOFs are appearing only in a few (2-3) MOFs. How the model behaves for predictions on materials containing metals that were not present in the training set

*) Given the relatively small size of the CoRE-MOF did the authors examined if their results are sensitive to the training-validation-test set random split?

*) Although in Fig. 3 the predicted isotherms for the MOF for the C₂H₂ adsorptions are compared very well to the experimentally determined ones, some more clarifications are still needed for the protocol used for the development of the model. More specifically:

i) During the training and validation of the predictive model, did some of the C₂H₂ adsorption capacities for the 3 MOFs at some pressures presented in Fig. 3(SIXSIX-2-Cu-i, Zn-MOF-74, ZJNU-103) were used for the training of the algorithm?

ii) In the SI in the Figures S20-S26, isotherms for the C₂H₂ adsorption for many materials are presented. The same in Figure S14-19 for the CO₂. My question: for a specific MOF and a specific gas, all points of the isotherms were predicted by the model (i.e., all points belong to the test set) or some of the isotherm points may belong to the training set ?

For me is very important to understand the previous point in order to assess the value of the proposed model. For example, if the entire isotherm for a MOF was predicted by the developed model then it may be not necessary for an experimentalist to synthesize a new material and study its properties. However, if "some" experimental information is needed for a specific MOF for the development of the predictive model, then someone should perform anyway experiments for this material. In the first case the value of the model is high, while in the second one is only limited.

Some points that need improvement:

*) No units are given anywhere for the MSE, RMSE, MAE etc. Providing just numbers is meaningless for evaluating the accuracy of a quantity. In general the authors do not mention the units of the adsorption capacities. (for example volumetric or gravimetric based units that are usually used or any other type)

*) In Tables S1-S10 is it necessary to include both MSE and its square root (RMSE)? One of the two is sufficient.

*) In SI it noted that the GCMC simulations were performed with the MS 2017R2 package. MS is

the Material Studio? Please provide the proper reference. Also, few lines after, what is 1x10⁷ steps?

*) Lines [144-147]: this part should be moved elsewhere, not in "Model performance and validation" section

*) There are some very strong statements in some places. I believe that some of them are not very accurate or they are oversimplifications (at least based on the present results) and should be rephrased/removed. In any case, as explained elsewhere there are additional works not reported in this manuscript. These statements are in lines [24-27], [49-53], [213-214]

*) There are several previous works in which CO₂ (and other gases) adsorptions were predicted by ML algorithms. For example by Fernandez (<https://doi.org/10.1021/acscombsci.5b00188>, <https://doi.org/10.1002/ejic.201600365>, <https://doi.org/10.1021/jp4006422>, <https://doi.org/10.1021/jz501331m>), and by Fanourgakis (<https://doi.org/10.1021/acs.jpcc.9b10766>) In some cases the reported value of R²=0.70 for CoRE-MOFs does not look very impressive.

Overall, I found the structure of the manuscript and the presentation of the results very confusing. I recognize however, that the authors were willing to provide to the readers all information needed.

For this reason their codes/data will become available. I am willing to review again this article, after the authors revise properly the manuscript and provide additional evidence for the value of their

approach that will help me to understand better. Along these lines I believe that: I) predictions for additional cases (gases, thermodynamic conditions etc) should be presented. In particular since there are several datasets available (e.g., hmofs-CH₄ etc), that could be directly used for the training/evaluation of their models, II) In the writing, I think most of the results (e.g., statistical metrics) should be tabulated in one place for the readers convenience and understanding). A so extensive discussion in the manuscript for various deep learning models I think that is unnecessary. Instead, previous efforts in the field should be discussed and compared with the present work.

Reviewer #2 (Remarks to the Author):

Following up on the previous reviewer comments and the reply from the authors, I have evaluated the work of Cui et al. and whether the questions raised by the reviewers were thoroughly addressed.

In general, I think the majority (although not all) of the comments and concerns raised by the reviewers were addressed. Additionally, I believe this work is suitable for publication in Nature Communications once the comments are addressed in full.

The biggest claim made in this work is that it has the "highest accuracy" of the available machine learning models. However, while new models were added in the revision, they are not MOF-specific models as was emphasized by Reviewer 1. Most of the models evaluated here, like LSTM, SOAP, and CGCNN, were not designed with MOFs in mind. Additionally, models like RACs, which have been used on MOFs, are not designed for this particular task, adsorption (it's not a surprise that a model based on RACs fails terribly because it is based solely on local and semi-local information; same with SOAP and several others). CGCNN is also 5 years old now; I would not consider it state-of-the-art anymore.

As cited by Reviewer 1, there are numerous MOF-focused ML papers to predict adsorption isotherms (here are a few notable additions:

<https://www.science.org/doi/10.1126/sciadv.abg3983>,

<https://pubs.acs.org/doi/10.1021/acs.jctc.9b00940>). In order to claim that the model has state-of-the-art performance, there needs to be a thorough and fair benchmark against existing state-

of-the-art models that are made for this task. I think this is currently missing.

On the topic of a fair benchmark, there are not enough details of the various models to reproduce the results and to determine if the benchmarking comparison is comparable. For instance, what hyperparameters were chosen for the competing models? For instance, there are numerous parameters in SOAP but none are described. The same goes for the other approaches like RACs and even MOF-specific models like MOFNet. Also, were the hyperparameters optimized for these models or used as-is? That is going to influence the validity of the model comparison process.

Finally, it's mentioned in the abstract that the model is even better than molecular simulations (i.e. GCMC), presumably when trained on experimental data. Can you elaborate a bit on why the molecular simulations are doing so poorly in Figure 3 compared to experiments? The disagreement between GCMC and the experiments is uncomfortably large, such that it begs the question why they are so off in the first place.

In addition, there are few very minor changes that should be made:

- Please include units on the MAE (e.g. in Figure 2 and elsewhere) and RMSE (e.g. in Figure 3 and elsewhere). Since it is an adsorption process, there should be units on the value.
- The authors cite Ref. 33 for the CoRE MOF Database, which is the 2014 paper, but they say the database contains over 11000 MOFs, which is only possible if the 2019 version is used. If the CoRE MOF 2019 database was used, please cite that and mention it in the text:
<https://pubs.acs.org/doi/10.1021/acs.jced.9b00835>.

Dear reviewers,

Thanks for your kind and constructive advice and comments to our manuscript entitled “Direct prediction of gas adsorption via spatial atom interaction learning” (NCOMMS-23-12213-T). We are pleased these constructive comments on our results. We have made modifications and corrections in response to these advice and comments. We provide clarifications on training method for the experimental database and details of the comparison methods, discussion of the performance for out-of-distribution metals and CO₂, CH₄ and H₂ predictions at higher pressure to fully assess the performance of the method. We believe that our revisions to the manuscript and SI, which are highlighted in yellow, have fully addressed the comments. Point-by-point changes and/or replies to the comments of referees are detailed as follows:

Comments from Referee 1:

In the present article Cui et al., presented an end-to-end deep learning model for accurate predictions of gas adsorption in nanoporous materials. According to the proposed approach predictive models of high accuracy have been developed requiring as an input only the structure of the material, namely the atomic coordinates of the chemical elements. It is claimed that the model outperforms any other model.

It is sad to say that after reading a few times the present manuscript, I still have not a safe opinion for the value of the present work. In particular the way that the predictive models were developed and evaluated still remains unclear to me despite the fact that several information was provided by the authors. More importantly, I am not sure what is the range of the applicability of the models developed.

Below are some of my concerns:

**) Why CO₂ and C₂H₂ adsorption were studied only at such low pressures (up to 1 bar)? There are freely available several datasets with the materials CO₂ capacities computed at higher pressures. Also, there are several datasets for different gases and at different thermodynamic conditions which could be used by the authors for the evaluation of the approach. Although it is explained the practical value of predicting CO₂ and C₂H₂ adsorption at the specific conditions, for the purpose of assessing the performance of a model additional results should be presented.*

It is known that from the physical point of view, there are different parameters that determine the adsorption of the gas at different thermodynamic conditions. For example, at low pressures the materials' composition is the most important factor, since the amount of the adsorbed gas is mainly determined by the strength of its interactions with the material. On the other hand, at high pressures the available volume is the most decisive factor. Examining the performance at a wide range of conditions is important for a thorough evaluation of the present model

Response: We thank the reviewer for the constructive comment. We have added the adsorption

capacity prediction performance of the model under high pressure for carbon dioxide (2.5 bar and 298 K), methane (35 bar and 298 K) and hydrogen (100 bar and 77 K), and compared it with other models. As presented in Supplementary Figure 28 and Supplementary Table 13, DeepSorption model still showed better performance compared with other models. The R^2 of the three tasks of CO_2 (2.5 bar and 298 K), CH_4 (35 bar and 298 K) and H_2 (100 bar and 77 K) adsorption reached 0.96, 0.98 and 0.99 respectively, higher than those adsorption task of low pressure, which may attribute to the fact that the adsorption capacity is mainly determined by the surface area and pore volume of the material under high pressure. This phenomenon can also be drawn from the better prediction effect of GEO_MLP on H_2 (100 bar and 77 K) task (R^2 : 0.994), than CGCNN (R^2 : 0.872), since the latter is not good at capturing the overall spatial structure of materials. At low pressure, the adsorption capacity of carbon dioxide is not only affected by the specific surface area and pore volume of the overall structure, but also by the chemical environment of pore channels such as local sites, so the prediction of its performance is more difficult. For C_2H_2 , due to the flammable and explosive characteristics of acetylene (explosion limit of 2.3%-72.3% in the air), the utilize and research pressure of C_2H_2 are generally under low pressure (0~1bar), so it is difficult to obtain enough data to train the model due to the rare studies on the adsorption of acetylene at high pressure for safety reasons.

Modifications: Supporting information, page 7:

We further examined the performance of DeepSorption model at a wide range of conditions, including carbon dioxide (2.5 bar and 298 K), methane (35 bar and 298 K) and hydrogen (100 bar and 77 K) adsorption capacity prediction tasks. As presented in Supplementary Figure 28 and Supplementary Table 13, DeepSorption model still showed better prediction performance compared with other models, and we also found that R^2 of adsorption prediction tasks of high pressure was generally higher than those of low pressure. The R^2 of the three tasks of CO_2 (2.5 bar and 298 K), CH_4 (35 bar and 298 K) and H_2 (100 bar and 77 K) adsorption reached 0.96, 0.98 and 0.99 respectively via DeepSorption models, which may attribute to the fact that the adsorption capacity is mainly determined by the surface area and pore volume of the material under high pressure. This phenomenon can also be drawn from the better prediction effect of GEO_MLP on H_2 (100 bar and 77 K) task (R^2 : 0.994), than CGCNN (R^2 : 0.872), since the latter is not good at capturing the overall spatial structure of materials.

Supporting information, page 61:

Task	Method	RMSE	MAE	R ²
H ₂ (100 bar and 77 K)	MatFormer+KCL	2.077	1.307	0.992
H ₂ (100 bar and 77 K)	MatFormer	2.220	1.408	0.991
H ₂ (100 bar and 77 K)	CGCNN	8.243	4.060	0.872
H ₂ (100 bar and 77 K)	GEO_MLP	1.772	0.897	0.994
CH ₄ (35 bar and 298 K)	MatFormer+KCL	0.581	0.450	0.985
CH ₄ (35 bar and 298 K)	MatFormer	0.584	0.443	0.985
CH ₄ (35 bar and 298 K)	CGCNN	1.004	0.677	0.955
CH ₄ (35 bar and 298 K)	GEO_MLP	1.209	0.803	0.934
CO ₂ (2.5 bar and 298 K)	MatFormer+KCL	0.582	0.425	0.956
CO ₂ (2.5 bar and 298 K)	MatFormer	0.625	0.454	0.949
CO ₂ (2.5 bar and 298 K)	CGCNN	1.257	0.848	0.796
CO ₂ (2.5 bar and 298 K)	GEO_MLP	1.334	0.928	0.769

Supplementary Table 13 | Prediction performance of adsorption capacity at high pressure in hMOF dataset. The calculated R² (Coefficient of determination), RMSE (Root Mean Square Error), MAE (Mean Absolute Error) of H₂, CH₄ and CO₂ adsorption capacity prediction on test set in hMOF dataset using different models to evaluate their prediction performance.

Supplementary Figure 28 | Prediction performance of adsorption capacity at high pressure in hMOF dataset. **a**, The correlations between true values and predicted values of H₂ adsorption capacity at 100 bar and 77 K on test set using DeepSorption (MatFormer+KCL) on hMOF dataset. **b**, The correlations between true values and predicted values of CH₄ adsorption capacity at 35 bar and 298 K on test set using DeepSorption on hMOF dataset. **c**, The correlations between true values

and predicted values of CO₂ adsorption capacity at 2.5 bar and 298 K on test set using DeepSorption on hMOF dataset. **d**, The correlations between true values and predicted values of H₂ adsorption capacity at 100 bar and 77 K on test set using MatFormer on hMOF dataset. **e**, The correlations between true values and predicted values of CH₄ adsorption capacity at 35 bar and 298 K on test set using MatFormer on hMOF dataset. **f**, The correlations between true values and predicted values of CO₂ adsorption capacity at 2.5 bar and 298 K on test set using MatFormer on hMOF dataset. **g**, The correlations between true values and predicted values of H₂ adsorption capacity at 100 bar and 77 K on test set using GEO_MLP on hMOF dataset. **h**, The correlations between true values and predicted values of CH₄ adsorption capacity at 35 bar and 298 K on test set using GEO_MLP on hMOF dataset. **i**, The correlations between true values and predicted values of CO₂ adsorption capacity at 2.5 bar and 298 K on test set using GEO_MLP on hMOF dataset. **j**, The correlations between true values and predicted values of H₂ adsorption capacity at 100 bar and 77 K on test set using CGCNN on hMOF dataset. **k**, The correlations between true values and predicted values of CH₄ adsorption capacity at 35 bar and 298 K on test set using CGCNN on hMOF dataset. **l**, The correlations between true values and predicted values of CO₂ adsorption capacity at 2.5 bar and 298 K on test set using CGCNN on hMOF dataset.

**) Some chemical elements (metals) out of the 77 in the Core MOFs are appearing only in a few (2-3) MOFs. How the model behaves for predictions on materials containing metals that were not present in the training set*

Response: We thank the reviewer for the constructive comment. The ability to predict out-of-distribution data is really challenging and we counted elements (As, Rh, Sb, Te, Ir, Pb, Np and Pu) that occur less than 10 times in the CoREMOf database for our study. As shown in Supplementary Figure 27, we have compared the predicted and true values of the CO₂ adsorption capacity of the MOF containing the rare element in the test set. DeepSorption still showed the best prediction performance (R^2 : 0.28), better than the other comparison models, CGCNN (R^2 :-0.23) RAC_MLP (R^2 :-0.15) and GEO_MLP (R^2 : 0.09). We attribute this good out-of-distribution prediction to the use of the Chemical Element Knowledge Graph method for element coding in MatFormer. (Molecular contrastive learning with chemical element knowledge graph, <https://ojs.aaai.org/index.php/AAAI/article/view/20313/20072>).

Modifications: Supporting information, page 6:

The ability to predict out-of-distribution data is challenging and is an important indicator of the model's generalizability. Elements that occur less than one in a thousand times are defined as rare elements (As, Rh, Sb, Te, Ir, Pb, Np and Pu). As shown in Supplementary Figure 27, we have compared the predicted and true values of the CO₂ adsorption capacity of the MOF containing the rare elements in the test set. DeepSorption still showed the best prediction performance (R^2 : 0.28), better than the other comparison models, CGCNN (R^2 :-0.23) RAC_MLP (R^2 :-0.15) and GEO_MLP (R^2 : 0.09). We attribute this good out-of-distribution prediction to the use of the Chemical Element Knowledge Graph for element coding in MatFormer. The Chemical Element Knowledge Graph coding method gives each element a vector representation containing chemical element information by learning and correlating the interrelationships between the properties of the elements. The vector representations are learnt based on Chemical Element Knowledge Graph and is independent to the training data of gas adsorption prediction, so that even if materials contain elements in the test data that do not appear in the training data or appear only a few times, the model can still give relatively accurate adsorption predictions since the out-of-distribution elements also have information-rich vector representations.

Supplementary Figure 27 | Adsorption capacity prediction performance of materials with rare elements in CoREMOF dataset. **a**, The correlations between true values and predicted values of CO₂ adsorption capacity of materials with rare elements (As, Rh, Sb, Te, Ir, Pb, Np and Pu) on test set with 10 different random divisions using CGCNN (**a**), RAC_MLP (**b**), GEO_MLP (**c**) and DeepSorption (MatFormer+KCL) (**d**) on CoREMOF dataset.

*) Given the relatively small size of the CoRE-MOF did the authors examined if their results are sensitive to the training-validation-test set random split?

Response: We thank the reviewer for the kind comment. Different splits of the dataset do have effects on the training and prediction results. To examine if the results are sensitive to the training-validation-test set random split, we performed 10 different random divisions of the CO₂ adsorption data in CoREMOF dataset and found that the results are less affected by the division of the dataset, with R² between 0.672-0.712 and MAE between 13.598-15.150, proving the robustness of the model.

Modifications: Supporting information, page 6:

To examine the sensitivity of DeepSorption towards the training-validation-test set random split of the prediction results, we performed 10 different random divisions of the CO₂ adsorption data in CoREMOF dataset (Supplementary Table 12) and found that the results are less affected by the division of the dataset, with R² between 0.672-0.712 and MAE between 13.598-15.150 cm³ g⁻¹, proving the robustness of the model.

Supporting information, page 60:

Experiment number	RMSE	MAE	R ²
1	20.519	14.050	0.706
2	21.614	14.192	0.693
3	20.801	14.275	0.698
4	21.189	13.730	0.698
5	21.568	14.203	0.704
6	20.720	13.598	0.712
7	22.444	15.150	0.672
8	21.461	14.708	0.691
9	21.791	15.067	0.701
10	20.206	14.057	0.705
mean	21.231	14.303	0.698
std	0.675	0.520	0.010

Supplementary Table 12 | Prediction performance of CO₂ adsorption capacity in CoREMOF dataset using DeepSorption. The calculated R² (Coefficient of determination), RMSE (Root Mean Square Error), MAE (Mean Absolute Error) of CO₂ adsorption capacity prediction on test set in CoREMOF dataset with different dataset divisions using DeepSorption models to evaluate their prediction performance.

**) Although in Fig. 3 the predicted isotherms for the MOF for the C₂H₂ adsorptions are compared very well to the experimentally determined ones, some more clarifications are still needed for the protocol used for the development of the model. More specifically:*

i) During the training and validation of the predictive model, did some of the C₂H₂ adsorption capacities for the 3 MOFs at some pressures presented in Fig. 3(SIXSIX-2-Cu-i, Zn-MOF-74, ZJNU-103) were used for the training of the algorithm?

ii) In the SI in the Figures S20-S26, isotherms for the C₂H₂ adsorption for many materials are presented. The same in Figure S14-19 for the CO₂. My question: for a specific MOF and a specific gas, all points of the isotherms were predicted by the model (i.e., all points belong to the test set) or some of the isotherm points may belong to the training set ?

For me is very important to understand the previous point in order to assess the value of the proposed model. For example, if the entire isotherm for a MOF was predicted by the developed model then it may be not necessary for an experimentalist to synthesize a new material and study its properties. However, if “some” experimental information is needed for a specific MOF for the development of the predictive model, then someone should perform anyway experiments for this material. In the first case the value of the model is high, while in the second one is only limited.

Response: We thank the reviewer for the kind comment. For a specific MOF and a specific gas, all points of the isotherms were predicted by the model and no isotherm points belong to the training set. We apologize for the misunderstanding caused by the lack of clarity in the experimental methods. We use leave-one-out method to evaluate the performance of our model in EXPMOF-CO₂ and EXPMOF-C₂H₂ datasets. When predicting the adsorption performance of material X (X refers to any material in the database), any adsorption data of material X will not appear in the training set for model training. We have added details to the test method on EXPMOF dataset to avoid misunderstandings to the readers.

Modifications: Main text, page 17:

We use leave-one-out method to evaluate the performance of our model in EXPMOF-CO₂ and EXPMOF-C₂H₂ datasets. In detail, when predicting the adsorption performance of material X (X refers to any material in the EXPMOF database) in the test set, any adsorption data of material X will not appear in the training set for model training.

Some points that need improvement:

*) No units are given anywhere for the MSE, RMSE, MAE etc. Providing just numbers is meaningless for evaluating the accuracy of a quantity. In general the authors do not mention the units of the adsorption capacities. (for example volumetric or gravimetric based units that are usually used or any other type)

Response: We thank the reviewer for the kind comment. We added the units of adsorption capacities to make it easier for readers to evaluate the performance of the model.

Modification: Main text, Page 8:

Fig. 2|Prediction performance of DeepSorption on CoREMOF and hMOF datasets.

Main text, Page 10:

Fig. 3|Prediction performance of DeepSorption on experimental dataset (EXP-MOF).

Extended Data Fig. 3|Prediction performance of DeepSorption on CoREMOF and hMOF datasets.

**) In Tables S1-S10 is it necessary to include both MSE and its square root (RMSE)? One of the two is sufficient.*

Response: We thank the reviewer for the kind comment. MSE can be converted to RMSE through simple mathematical methods (square root), so we deleted MSE in Tables S1-S13.

**) In SI it noted that the GCMC simulations were performed with the MS 2017R2 package. MS is the Material Studio? Please provide the proper reference. Also, few lines after, what is 1×10^7 steps?*

Response: We thank the reviewer for the kind comment. We are sorry for not giving the full name of the software and for omitting the superscript of the index numbers, which caused a misunderstanding. As the reviewer said, MS is Material Studio software and 1×10^7 should be 1×10^7 . We have corrected and added these in the supporting information accordingly.

Modifications: Supporting information, page 12:

All the GCMC simulations were performed in Material Studio 2017R2 package.

The loading steps and the equilibration steps were 1×10^7 , the production steps were 1×10^7 .

**) Lines [144-147]: this part should be moved elsewhere, not in “Model performance and validation” section*

Response: We thank the reviewer for the kind comment. Based on the reviewer's comments, we moved lines [144-147] to supporting information and made contextual changes to make it more logically coherent.

Modifications: Supporting information, Page 5:

Considering the importance of direct air capture and acetylene, methane and hydrogen storage technologies, CoREMOF (CO₂), hMOF (CO₂, N₂, H₂, CH₄) and EXPMOF (CO₂, C₂H₂) datasets were utilized to train the models.

**) There are some very strong statements in some places. I believe that some of them are not very accurate or they are oversimplifications (at least based on the present results) and should be rephrased/removed. In any case, as explained else where there are additional works not reported in this manuscript. These statements are in lines [24-27], [49-53], [213-214]*

Response: We thank the reviewer for the kind comment. Based on the reviewers' comments, we rephrased some statements to make the description more accurate.

Main text, page 2

Complete adsorption curves prediction could be performed using DeepSorption with a higher accuracy than molecular simulations and other machine learning models, a 20-35% decline in the mean absolute error compared to current state-of-the-art models.

Main text, page 3

However, the accurate prediction of adsorption performance still remains a challenge due to the complex associations between raw material structures and functional properties, which requires

machine learning models to understand the correlations among global atoms, local atoms with different element definitions.

Main text, page 11

Despite a longer computing time (tens of hours), the adsorption prediction performance of molecular simulation is still unsatisfactory in the low pressure adsorption zone (Fig. 3c-3e).

**) There are several previous works in which CO₂ (and other gases) adsorptions were predicted by ML algorithms. For example by Fernandez (<https://doi.org/10.1021/acscmbosci.5b00188>, <https://doi.org/10.1002/ejic.201600365> , <https://doi.org/10.1021/jp4006422> ,*

<https://doi.org/10.1021/jz501331m>), and by Fanourgakis (<https://doi.org/10.1021/acs.jpcc.9b10766>) In some cases the reported value of R²=0.70 for CoRE-MOFs does not look very impressive.

Overall, I found the structure of the manuscript and the presentation of the results very confusing. I recognize however, that the authors were willing to provide to the readers all information needed. For this reason their codes/data will become available. I am willing to review again this article, after the authors revise properly the manuscript and provide additional evidence for the value of their approach that will help me to understand better. Along these lines I believe that: I) predictions for additional cases (gases, thermodynamic conditions etc) should be presented. In particular since there are several datasets available (e.g., hmoFs-CH₄ etc), that could be directly used for the training/evaluation of their models, II) In the writing, I think most of the results (e.g., statistical metrics) should be tabulated in one place for the readers convenience and understanding). A so extensive discussion in the manuscript for various deep learning models I think that is unnecessary. Instead, previous efforts in the field should be discussed and compared with the present work.

Response: We thank the reviewer for the kind comment. Since the prediction difficulty of different adsorption tasks varies greatly, the prediction performance of the same model may vary greatly for different material database or for different gases. To evaluate the performance of the model reasonably, different algorithm models are used to deal with the same task. As presented in supplementary Table 1-3 and 13, DeepSorption shows better performance than other models in complex CO₂ tasks. According to the reviewer's opinions, we also carried out the prediction of methane and hydrogen under high pressure of hMOF database, and found that DeepSorption model also achieved excellent prediction performance compared with other models (Supplementary Table 1, 2, 3 and 13). However, since the task is relatively simple, all the employed models show fairly satisfied performance. The detailed explanation has been supplied in Supporting Information.

Modifications: Supporting information, page 7:

We further examined the performance of DeepSorption model at a wide range of conditions, including carbon dioxide (2.5 bar and 298 K), methane (35 bar and 298 K) and hydrogen (100 bar and 77 K) adsorption capacity prediction tasks. As presented in Supplementary Figure 28 and Supplementary Table 13, DeepSorption model still showed better prediction performance compared with other models, and we also found that R² of adsorption prediction tasks of high

pressure was generally higher than those of low pressure. The R^2 of the three tasks of CO_2 (2.5 bar and 298 K), CH_4 (35 bar and 298 K) and H_2 (100 bar and 77 K) adsorption reached 0.96, 0.98 and 0.99 respectively via DeepSorption models, which may attribute to the fact that the adsorption capacity is mainly determined by the surface area and pore volume of the material under high pressure. This phenomenon can also be drawn from the better prediction effect of GEO_MLP on H_2 (100 bar and 77 K) task (R^2 : 0.994), than CGCNN (R^2 : 0.872), since the latter is not good at capturing the overall spatial structure of materials.

Supporting information, page 61:

Task	Method	RMSE	MAE	R^2
H_2 (100 bar and 77 K)	MatFormer+KCL	2.077	1.307	0.992
H_2 (100 bar and 77 K)	MatFormer	2.220	1.408	0.991
H_2 (100 bar and 77 K)	CGCNN	8.243	4.060	0.872
H_2 (100 bar and 77 K)	GEO_MLP	1.772	0.897	0.994
CH_4 (35 bar and 298 K)	MatFormer+KCL	0.581	0.450	0.985
CH_4 (35 bar and 298 K)	MatFormer	0.584	0.443	0.985
CH_4 (35 bar and 298 K)	CGCNN	1.004	0.677	0.955
CH_4 (35 bar and 298 K)	GEO_MLP	1.209	0.803	0.934
CO_2 (2.5 bar and 298 K)	MatFormer+KCL	0.582	0.425	0.956
CO_2 (2.5 bar and 298 K)	MatFormer	0.625	0.454	0.949
CO_2 (2.5 bar and 298 K)	CGCNN	1.257	0.848	0.796
CO_2 (2.5 bar and 298 K)	GEO_MLP	1.334	0.928	0.769

Supplementary Table 13 | Prediction performance of adsorption capacity at high pressure in hMOF dataset. The calculated R^2 (Coefficient of determination), RMSE (Root Mean Square Error), MAE (Mean Absolute Error) of H_2 , CH_4 and CO_2 adsorption capacity prediction on test set in hMOF dataset using different models to evaluate their prediction performance.

Comments from Referee 2:

Following up on the previous reviewer comments and the reply from the authors, I have evaluated the work of Cui et al. and whether the questions raised by the reviewers were thoroughly addressed.

In general, I think the majority (although not all) of the comments and concerns raised by the reviewers were addressed. Additionally, I believe this work is suitable for publication in Nature Communications once the comments are addressed in full.

The biggest claim made in this work is that it has the “highest accuracy” of the available machine learning models. However, while new models were added in the revision, they are not MOF-specific models as was emphasized by Reviewer 1. Most of the models evaluated here, like LSTM, SOAP, and CGCNN, were not designed with MOFs in mind. Additionally, models like RACs, which have been used on MOFs, are not designed for this particular task, adsorption (it’s not a surprise that a model based on RACs fails terribly because it is based solely on local and semi-local information; same with SOAP and several others). CGCNN is also 5 years old now; I would not consider it state-of-the-art anymore.

As cited by Reviewer 1, there are numerous MOF-focused ML papers to predict adsorption isotherms (here are a few notable additions: <https://www.science.org/doi/10.1126/sciadv.abg3983>, <https://pubs.acs.org/doi/10.1021/acs.jctc.9b00940>). In order to claim that the model has state-of-the-art performance, there needs to be a thorough and fair benchmark against existing state-of-the-art models that are made for this task. I think this is currently missing.

On the topic of a fair benchmark, there are not enough details of the various models to reproduce the results and to determine if the benchmarking comparison is comparable. For instance, what hyperparameters were chosen for the competing models? For instance, there are numerous parameters in SOAP but none are described. The same goes for the other approaches like RACs and even MOF-specific models like MOFNet. Also, were the hyperparameters optimized for these models or used as-is? That is going to influence the validity of the model comparison process.

Response: We thank the reviewer for the kind comments. We have added details of the various methods, including SOAP and MOFNet hyperparameters, so that readers can better reproduce them. And as presented in Figs. 2e, 2f and supplementary Table 13 Supplementary Table 13, our model DeepSorption achieved the best prediction results for carbon dioxide, nitrogen, methane, and hydrogen in both CoREMOF and hMOF databases, and the used framework and hyperparameters are adopted from the article code. It is worth mentioning that MOFNet is also a highly universal framework for MOF adsorption performance prediction published in November last year, and has achieved good prediction performance. At the same time, a fair comparison between different models is complex, because of the different models tend to use different input data to solve different problems. For example, in this article <https://www.science.org/doi/10.1126/sciadv.abg3983>, it is to predict the unknown adsorption amount of the same material under other pressures and temperatures by the known adsorption amount under known pressure and temperature. It does not use the structure information of the

porous material itself as an input, but uses its temperature, pressure and the corresponding adsorption amount under the working condition as an input. The method in the article of <https://pubs.acs.org/doi/10.1021/acs.jctc.9b00940> is limited by using the molecular composition of ligand and extract its functional groups to input, although such methods have achieved good prediction results in the adsorption prediction at high pressure, the universality of the method is compromised (theoretically, the types of metal cluster nodes and ligands are infinite). The goal of our work is not only the accuracy of prediction, but also the universality of the model, the simplicity of the input method and the end-to-end prediction mode. In theory, the input of DeepSorption model for adsorption performance prediction (atomic coordinates and corresponding element types and cell parameters) can be directly obtained for any crystal material. On the basis of the above, we use machine learning models based on SOAP, RAC, GEO descriptors and CGCNN, which also have high universality of machine learning models, for performance comparison.

Modifications: Supporting information, page 12:

MOFNet²⁵ is an interpretable graph transformer network for predicting adsorption isotherms of MOFs. The prediction of MOFNet originates from both local representation and global representation. The number of graph transformer network layers was set to 2 and the hidden state of each layer was set to 1024. We applied a three-layer MLP as the global feature encoder, and the dimensions of these three layers were set to 128, 512, and 1024. The model was trained for 300 epoch using a batch size of 32 MOF structures. We split the datasets with a ratio for train/validation/test as 0.7:0.15:0.15 in the CoREMOF dataset. We trained all codes on a Ubuntu Server with 1 GPUs (NVIDIA GeForce 3090Ti). MOFNet is trained to minimize the MSE loss, which is the mean overseen data of the squared differences between true and predicted values. We used the Adam optimizer with $\beta_1 = 0.9$, $\beta_2 = 0.98$, and $\epsilon = 10^{-9}$.

Supporting information, page 13:

For models based on geometric models, the descriptors are LCD (largest cavity diameter), PLD (pore limiting diameter), D (density), ASA (accessible surface area), VF (void fraction), AV (accessible volume). For hMOF dataset, the descriptors are LCD (largest cavity diameter), PLD (pore limiting diameter), D (density), ASA (accessible surface area), VF (void fraction), which are calculated using Zeo⁺⁺ programs with radius_of_area_probe of 1.655, area_monte_carlo_samples of 2000 and porosity_monte_carlo_samples of 100000. The RACs descriptors are calculated using Molsimplify packages³¹. The MBTR descriptors are calculated using MBTR package²⁹. The SOAP descriptors are calculated using DScribe package³² with rcut of 8, nmax of 8, lmax of 6, σ of 0.2 and average method of inner average.

Finally, it's mentioned in the abstract that the model is even better than molecular simulations (i.e. GCMC), presumably when trained on experimental data. Can you elaborate a bit on why the molecular simulations are doing so poorly in Figure 3 compared to experiments? The disagreement between GCMC and the experiments is uncomfortably large, such that it begs the question why they are so off in the first place.

Response: We thank the reviewer for the kind comment. The simulated values through GCMC for weakly polar gases such as nitrogen, hydrogen and methane are in good agreement with the experimental values, but for strongly polar gases such as alkynes and olefin gases, the simulated values differ greatly from the experimental values. Despite the importance of GCMC in high-throughput screening, the deviations between GCMC and experimental adsorption data are long suffered, especially in the case of porous materials with flexibility and strong binding sites, which are difficult to be precisely described using force field. (Evaluation of Force Field Performance for High-Throughput Screening of Gas Uptake in Metal–Organic Frameworks, <https://pubs.acs.org/doi/pdf/10.1021/jp511674w>, Screening of Metal–Organic Frameworks for Carbon Dioxide Capture from Flue Gas Using a Combined Experimental and Modeling Approach, <https://pubs.acs.org/doi/pdf/10.1021/ja9057234>)

The materials selected in the maintext are SIFSIX-2-Cu-i (fluorine anions), Zn-MOF-74 (open metal sites) and ZJNU-103 (Amino functional groups) with strong adsorption sites and flexible framework, which is difficult to be accurately simulated using the generic MOF force field. In previous report, there have been cases where the experimental and high-throughput calculations for this type of materials deviated, for example, ZU-62 (also known as NbOFFIVE-2-Cu-i) showed an experimental Xenon adsorption of 3.21 mmol g⁻¹ at 1 bar and 298 K with a Xenon/Krypton selectivity of 9.7 (Separation of Xe from Kr with Record Selectivity and Productivity in Anion-Pillared Ultramicroporous Materials by Inverse Size-Sieving, <https://onlinelibrary.wiley.com/doi/epdf/10.1002/ange.201913245>), whereas in the GCMC simulations Xenon adsorption was only 0.01 mmol g⁻¹ with a Xenon/Krypton selectivity of 0.03 (Construction of an Anion-Pillared MOF Database and the Screening of MOFs Suitable for Xe/Kr Separation, <https://pubs.acs.org/doi/pdf/10.1021/acsami.1c00152>). This deviation is also present in the GCMC simulations for acetylene, carbon dioxide and ethylene. (Machine-learning-assisted exploration of anion-pillared metal organic frameworks for gas separation, <https://doi.org/10.1016/j.matt.2022.07.029>) The AI approach, on the other hand, predicts the data by using the model learned from the experimental data and therefore allows better learning of the flexibility and strong binding sites of the material. This is the advantage of the AI approach over high-throughput simulation methods, not only in terms of higher prediction efficiency, but also in terms of higher prediction accuracy when based on experimental data as a standard and trained with experimental data.

Modifications: Supporting information, page 13:

It's worth noting that the simulated values through GCMC for weakly polar gases such as nitrogen, hydrogen and methane are in good agreement with the experimental values, but for strongly polar gases such as alkynes and olefin gases, the simulated values may differ greatly from the experimental values.

In addition, there are few very minor changes that should be made:

1. Please include units on the MAE (e.g. in Figure 2 and elsewhere) and RMSE (e.g. in Figure 3 and elsewhere). Since it is an adsorption process, there should be units on the value.

Response: We thank the reviewer for the kind comment. We added the units of adsorption capacities to make it easier for readers to evaluate the performance of the model.

Modification: Main text, Page 8:

Fig. 2|Prediction performance of DeepSorption on CoREMOF and hMOF datasets.

Main text, Page 10:

Fig. 3|Prediction performance of DeepSorption on experimental dataset (EXP-MOF).

Extended Data Fig. 3|Prediction performance of DeepSorption on CoREMOF and hMOF datasets.

2. The authors cite Ref. 33 for the CoRE MOF Database, which is the 2014 paper, but they say the database contains over 11000 MOFs, which is only possible if the 2019 version is used. If the CoRE MOF 2019 database was used, please cite that and mention it in the text: <https://pubs.acs.org/doi/10.1021/acs.jced.9b00835>.

Response: We thank the reviewer for the constructive comment. We have changed the referenced articles of the database so that readers can access the corresponding database directly.

Modification: Maintext, Page 21:

33 Chung, Y. G. et al. Advances, updates, and analytics for the Computation-Ready, Experimental Metal–Organic Framework database: CoRE MOF 2019. *J. Chem. Eng. Data* **64**, 5985-5998(2019).

We believe that we have satisfactorily addressed all comments and that the manuscript is thereby improved. We appreciate your suggestions for improvements to our manuscript.
Best regards,

Huabin Xing, Ph.D.
Qishi Distinguished Professor
College of Chemical and Biological Engineering
Zhejiang University, Hangzhou 310027, China

REVIEWER COMMENTS

Reviewer #1 (Remarks to the Author):

After carefully reviewing the replies of the authors to the reviewer comments, I believe that all my concerns were properly addressed. The manuscript "Direct prediction of gas adsorption via spatial atom interaction learning" is a significant contribution to the field and deserves publication in the Nature Communications.

Reviewer #2 (Remarks to the Author):

The authors have addressed most of my comments. My remaining concerns have to do with the language of some of the claims in places. For instance:

"Even compared with the current state-of-the-art graph neural 187 networks, such as CGCNN...". As noted in my prior review, CGCNN was one of the first graph convolutional neural nets for materials, and there are many better models than it in the literature. DimeNet++ is one of many such examples (there are dozens). It's fine to compare to CGCNN, but a claim that the model is better than state-of-the-art graph neural nets can't be made in the manuscript's current state. The claim should either be made to be more representative of the submitted work or other models need to be tested.

Also, in the abstract it states "a 20-35% decline in the 27 mean absolute error compared to current state-of-the-art models." In my prior review, I stated "It's not a surprise that a model based on RACs fails terribly because it is based solely on local and semi-local information; same with SOAP and several others." I still believe this to be the case. It's fine to state that the proposed model performs better, but I would recommend shifting the focus away from "state-of-the-art models." Perhaps "compared to more general-purpose models that take only a crystal structure as input" would be more suitable.

Please also include the versions of Zeo++ and DSCRIBE used in this work. Mentioning the version for the latter is particularly important since different versions of DSCRIBE implement SOAP slightly differently.

Otherwise, I believe that my comments have been addressed. That said, I agree with many of Reviewer 1's comments but will defer to their judgment about whether they have been satisfactorily addressed in this revision.

Dear reviewers,

Thanks for your kind and constructive comments to our manuscript entitled “Direct prediction of gas adsorption via spatial atom interaction learning” (NCOMMS-23-12213A). We are pleased about these kind and constructive comments on our results. We have made careful modifications and corrections in response to these advice and comments. We have changed some of the description details in the article to make it more accurate and clear, and provided versions of the data processing software used in the study so that readers can better reproduce our work. We believe that our revisions to the manuscript and SI, which are highlighted in yellow, have fully addressed the comments. Point-by-point changes and/or replies to the comments of referees are detailed as follows:

Comments from Referee 1:

After carefully reviewing the replies of the authors to the reviewer comments, I believe that all my concerns were properly addressed. The manuscript “Direct prediction of gas adsorption via spatial atom interaction learning” is a significant contribution to the field and deserves publication in the Nature Communications.

Response: Thank you very much for your comments, and thank you again for your previous constructive suggestions to improve our work.

Comments from Referee 2:

The authors have addressed most of my comments. My remaining concerns have to do with the language of some of the claims in places. For instance:

"Even compared with the current state-of-the-art graph neural 187 networks, such as CGCNN...". As noted in my prior review, CGCNN was one of the first graph convolutional neural nets for materials, and there are many better models than it in the literature. DimeNet++ is one of many such examples (there are dozens). It's fine to compare to CGCNN, but a claim that the model is better than state-of-the-art graph neural nets can't be made in the manuscript's current state. The claim should either be made to be more representative of the submitted work or other models need to be tested.

Also, in the abstract it states "a 20-35% decline in the 27 mean absolute error compared to current state-of-the-art models." In my prior review, I stated "It's not a surprise that a model based on RACs fails terribly because it is based solely on local and semi-local information; same with SOAP and several others." I still believe this to be the case. It's fine to state that the proposed model performs better, but I would recommend shifting the focus away from "state-of-the-art models." Perhaps "compared to more general-purpose models that take only a crystal structure as input" would be more suitable.

Response: We thank the reviewer for the kind comment. We have changed these description details in the article to make it more accurate and clear. Just like the reviewer said, CGCNN is the first and most representative graph neural networks for material representation learning. In the previous revision, we have compared our model with the CGCNN model and the MOFNet model (a model based on graph neural network for MOFs adsorption capacity prediction) on the adsorption performance prediction task. Because of the different downstream tasks, the input of different graph neural networks may be different and is not necessarily suitable for the downstream tasks of adsorption performance prediction. Meanwhile, considering that there are too many models based on graph neural networks, it is not possible to make a full comparison, so we have modified our statement to make it clearer and more accurate.

Modifications: Main text, page 9:

Compared with the graph neural network CGCNN, the performance is also significantly improved, from 0.48 to 0.70 of R^2 (Fig. 2e).

Modifications: Main text, page 2:

....., a 20-35% decline in the mean absolute error compared to graph neural network CGCNN and machine learning models based on descriptors.

Please also include the versions of Zeo++ and DScibe used in this work. Mentioning the version for the latter is particularly important since different versions of DScibe implement SOAP slightly differently.

Otherwise, I believe that my comments have been addressed. That said, I agree with many of Reviewer 1's comments but will defer to their judgment about whether they have been satisfactorily addressed in this revision.

Response: We thank the reviewer for the kind comment. We have added versions of the software used in this research, including Zeo++, Molsimplify, MBTR and DScibe, so that readers can better reproduce them.

Modification: Supporting information, Page 13:

For hMOF dataset, the descriptors are LCD (largest cavity diameter), PLD (pore limiting diameter), D (density), ASA (accessible surface area), VF (void fraction), which are calculated using Zeo++ programs (version: 0.3) with radius_of_area_probe of 1.655, area_monte_carlo_samples of 2000 and porosity_monte_carlo_samples of 100000. The RACs descriptors are calculated using Molsimplify packages³¹(version: 1.7.2). The MBTR descriptors are calculated using MBTR package²⁹ (version: 0.0.1). The SOAP descriptors are calculated using DScibe package³² (version: 2.0.1) with rcut of 8, nmax of 8, lmax of 6, σ of 0.2 and average method of inner average.

We believe that we have satisfactorily addressed all comments and that the manuscript is thereby improved. We appreciate your suggestions for improvements to our manuscript.
Best regards,

Huabin Xing, Ph.D.
Qishi Distinguished Professor
College of Chemical and Biological Engineering
Zhejiang University, Hangzhou 310027, China